# Endothelial Cell Behavior and Nitric Oxide Production on a-C:H:SiO_x_-Coated Ti-6Al-4V Substrate

**DOI:** 10.3390/ijms24076675

**Published:** 2023-04-03

**Authors:** Igor A. Khlusov, Alexander S. Grenadyorov, Andrey A. Solovyev, Vyacheslav A. Semenov, Maksim O. Zhulkov, Dmitry A. Sirota, Aleksander M. Chernyavskiy, Olga V. Poveshchenko, Maria A. Surovtseva, Irina I. Kim, Natalya A. Bondarenko, Viktor O. Semin

**Affiliations:** 1Laboratory of Cellular and Microfluidic Technologies, Siberian State Medical University, 2, Moskovskii Tract, 634050 Tomsk, Russia; 2The Institute of High Current Electronics SB RAS, 2/3, Akademichesky Ave., 634055 Tomsk, Russia; 3E.N. Meshalkin National Medical Research Center of Ministry of Health of Russian Federation, 15, Rechkunovskaya Str., 630055 Novosibirsk, Russia; 4Research Institute of Clinical and Experimental Lymphology, Branch of Institute of Cytology and Genetics SB RAS, 2, Timakov Str., 630060 Novosibirsk, Russia; 5Institute of Strength Physics and Materials Science SB RAS, 2/4, Akademichesky Ave., 634055 Tomsk, Russia

**Keywords:** a-C:H:SiO_x_ coating, local atomic order, endothelial cells EA.hy926, in vitro cytotoxicity, adhesion, cell number, nitric oxide production

## Abstract

This paper focuses on the surface modification of the Ti-6Al-4V alloy substrate via a-C:H:SiO_x_ coating deposition. Research results concern the a-C:H:SiO_x_ coating structure, investigated using transmission electron microscopy and in vitro endothelization to study the coating. Based on the analysis of the atomic radial distribution function, a model is proposed for the atomic short-range order structure of the a-C:H:SiO_x_ coating, and chemical bonds (C–O, C–C, Si–C, Si–O, and Si–Si) are identified. It is shown that the a-C:H:SiO_x_ coating does not possess prolonged cytotoxicity in relation to EA.hy926 endothelial cells. In vitro investigations showed that the adhesion, cell number, and nitric oxide production by EA.hy926 endothelial cells on the a-C:H:SiO_x_-coated Ti-6Al-4V substrate are significantly lower than those on the uncoated surface. The findings suggest that the a-C:H:SiO_x_ coating can reduce the risk of endothelial cell hyperproliferation on implants and medical devices, including mechanical prosthetic heart valves, endovascular stents, and mechanical circulatory support devices.

## 1. Introduction

Over the past 15 years, heart failure has remained a leading cause of death throughout the world [1,2]. In developed countries, the heart failure prevalence rate is between 1 and 2% of the total adult population [3,4,5]. The heart failure prevalence rate is predicted to be 46% higher by 2030 and, consequently, will continue to be the main cause of death from cardiovascular disease [6].

Coronary heart disease and cardiac valve damage take leading places among other causes of chronic heart failure [1]. Coronary artery stenting is the basic prevention technique from the progression of cardiac insufficiency of patients with coronary heart disease. However, the problems of the implant’s biointegration and the high risk of artery restenosis development after stenting remain essentially problematic.

The development and application of intracoronary drug-eluting stents (DES), have become a turning point in interventional cardiology practice, enabling a reduction in mortality and early artery restenosis after coronary stent implantation, from 30 to 10%, in using bare metal, respectively [7]. Despite technological progress, the useful DES effect decreases progressively as the drug and its delivery system dissolve. In this case, the metal base baring considerably increases the risk of delayed restenosis due to the excessive proliferation (hyperplasia) of the intima cells of blood vessels [8,9], or thrombosis of endovascular stents due to the drug-driven endothelization inhibition and prolonged endothelial dysfunction during interventional treatment of occlusive cardiovascular diseases.

Moreover, genetic mutations developing over time in endothelial cells lead to a decrease in their sensitivity and a higher resistance to cytostatic agents (e.g., rapamycin) of drug-eluting coatings [10,11]. As a consequence, the risk of bare metal stent and DES restenosis ranges within 16–44% and 3–20%, respectively [12]. Therefore, the long-term implant patency and functionality correlate with its surface biocompatibility, enabling a reduction in the risk of thrombosis and/or restenosis [13,14].

At present, there are a variety of prostheses for transcatheter replacement of aortic, tricuspid, and mitral valves. All of these valves have rather vast bare metal regions freely contacting blood [15]. Implantation of a heart pump, e.g., a left ventricular assist device, is the first stage of heart transplantation, which is used to support patients at the terminal stage of chronic heart failure. One of the main weaknesses of implanting mechanical heart valves and pumps is the risk of thromboembolic disorders [16,17].

The material used for prosthesis fabrication can activate the blood coagulation system and blood complement, thereby provoking thrombus formation [18]. Moreover, the blood stream flowing through mechanical devices can induce high shear stress, which causes red blood cell (RBC) hemolysis. Disturbances of the blood stream laminarity in the mechanical device can also initiate platelet aggregation, leading to thrombus formation, with a critical effect on the patient [19,20]. Therefore, the surface of cardiovascular implants and devices must possess low thrombogenicity. Both in theory and experimentally, endothelization of the artificial surface decreases the thrombocytic activity in blood coagulation tests [21].

However, the complete endothelization of the blood-contacting surfaces of cardiovascular devices (e.g., cameras and moving parts) is not always optimal for preventing post-implantation complications. Many complex devices are needed to prevent the possibility of growing additional tissues and objects between moving parts of cardiovascular device. For example, circulatory support systems, despite their large contact area with blood and the high risk of thromboembolic complications, do not allow for implementation of the concept of biointegrated systems through surface encapsulation/complete neo-endothelization because of strict requirements for space between rotating elements and heart pump housing.

A similar problem arises in the case with artificial heart valves, the important assembly of which is the fixation system of the locking mechanism. In this case, in addition to drug therapy, the problem of prevention of thromboembolic complications can be solved with the development of hypothrombogenic coatings, which hinder active cell adhesion and migration. The developed a-C:H:SiO_x_ coating significantly decreases in vitro platelet adhesion [22,23]. The evident benefit of such a coating applied to cardiovascular implants will have a prolonged antiproliferative effect in relation to endothelial cells.

Biodegradable polymer coatings are used to improve the surface biocompatibility of stents and drug delivery, but can induce the development of the inflammatory reaction, which is supposed to be the basis of neointimal hyperplasia [24]. At the same time, the use of stable coatings helps to avoid the inflammatory reaction of endotheliocytes, thanks to the protective barrier impeding the release of toxic chemicals from substrates [25,26].

One of the most proposed solutions of biocompatibility is a deposition of thin diamond-like coatings (DLCs). From the start of the 2000s, it has been shown that a DLC is bioinert, resistant to mechanical loads and corrosion, and not cytotoxic in relation to monocytes/macrophages, fibroblasts, or osteoblasts [27]. In virtue of the optimum ratio of *sp*^2^–*sp*^3^ carbon atoms, this coating possesses good hemocompatibility [28,29]. Owing to a certain dissatisfaction with biomedical DLC testing, research on its physicochemical modification with silicon, and silicon oxides in particular, has become very popular in the past 5 years, because it improves the surface properties of medical materials and devices [30]. Our in vitro investigations [31] show an absence of cytotoxicity in hydrogenated carbon coatings with silicon and oxygen (a-C:H:SiO_x_) in relation to leukocytes, retardation of pro-inflammatory cytokine/chemokine secretion, prevention of RBC hemolysis [32], and platelet adhesion [22] in combination with the advanced mechanical, corrosion-resisting, and tribological properties of thin a-C:H:SiO_x_ coatings [22,31,32].

The aim of this work is to study the atomic structure of the a-C:H:SiO_x_ coating deposited onto the Ti-6Al-4V substrate, and to analyze in vitro investigations of the interaction between endothelial cells and a-C:H:SiO_x_-coated substrates.

## 2. Results and Discussion

### 2.1. Atomic Structure of the a-C:H:SiO_x_ Coating

Our previous research identified the effects of the argon pressure, substrate bias voltage, and distance between the plasma generator and substrate, as well as, magnetic field, and polyphenyl methylsiloxane flow rate on the structure and properties of the a-C:H:SiO_x_ coating [33,34,35,36,37,38]. These dependences helped to optimize the a-C:H:SiO_x_ coating deposition process in order to acquire high hardness, a low friction coefficient, a low wear rate, and appropriate corrosion resistance [23,31,32].

The interaction between implants and the bodily fluid is strongly influenced by their surface properties, which directly affect various postimplant biological reactions, including precipitation of different minerals, protein adsorption, cell adhesion, and proliferation [38]. Bociaga et al. [39] report that the optimum silicon content for exerting a positive effect on medical and biological properties is 10–14 at.%. A silicon content in the coating higher than 20 at.%, results in chemical and structural changes, reducing its biocompatibility.

According to the EDX analysis, the optimum conditions of the a-C:H:SiO_x_ coating deposition provide 85 at.% C, 11 at.% Si, and ~4 at.% O. The SEM image of the a-C:H:SiO_x_-coated substrate surface is presented in Figure 1.

The bright-field TEM image presented in Figure 2a shows the cross-sectional view of the titanium alloy with the deposited a-C:H:SiO_x_ coating. One can observe that the coating structure is amorphous. The respective NBD pattern in Figure 2b demonstrates only an expanded area of the diffraction scattering. The coating/substrate interface is well defined. A series of three NBD patterns is used to construct the function of the radial distribution function (RDF) of atoms in the coating. This function links the experimentally measured diffraction-scattering intensity of the amorphous phase and the atom distribution in the *n*-th coordination shell.

In Figure 3, RDF peaks matching the first and second coordination shells are resolved into pair atomic spacings using the Gaussian function. The atomic short-range order is well pronounced in the coating, which is restricted by two coordination shells. Coordination numbers determined as areas under peaks are 7 and 14 for the first and second coordination shells, respectively. In the first coordination shell, there are two atomic spacings centered on 1.13–1.22 Å and 1.33–1.42 Å, which corresponds to C–O and C–C bonds, respectively. In the second coordination shell, there are at least four peaks corresponding to 2.0–2.12 Å, 2.23–2.37 Å, 2.59–2.73 Å, and 2.83–3.01 Å atomic spacings, matching Si–C (Si–O), C–C, Si–Si, and C–C bonds, respectively. The type of chemical bond is identified through the comparison of radii experimentally obtained for coordination shells and atoms according to Goldschmidt [40], i.e., *r*_C_ = 0.77 Å, *r*_O_ = 0.60 Å, *r*_Si_ = 1.17 Å.

Our results are in good agreement with those obtained by Hilbert et al. [41], who report on the presence of short-range order atomic clusters in the a-C:H:SiO_x_ coating with the DLC structure. Carbon *sp*^3^ hybridization occurs in these clusters. Based on the length of C–Si and C–C bonds (Figure 3) and the distribution of *sp*^3^ hybridized bonds in C–H–Si–O molecules [41], it is possible to identify the formation of the atomic short-range order in the synthesized a-C:H:SiO_x_ coating, which is also present in the DLC structure with the dominant nonpolar and polar covalent bonds C–C and Si–C(O), respectively. TEM results are consistent with the Fourier transform infrared analysis obtained in our previous research [42], and confirm the presence of C–C, Si–O, Si–C, and C–O bonds in the coating.

Based on the RDF results, a model of the short-range order atomic structure is proposed for the a-C:H:SiO_x_ coating. The structure consists of clusters based on carbon, oxygen, and silicon atoms. This model is presented in Figure 4.

As applied to medicine, the sign of the electric charge and the amplitude of the electrokinetic potential (zeta-potential), associated with the Coulomb potential, are the key parameters of the a-C:H:SiO_x_ coating. These parameters must be controlled, since implants contact hemocytes possessing the negative zeta-potential [43]. In the case of negative zeta-potential of the implant surface, the risk of direct hemocyte adhesion to the artificial surface considerably lowers.

As shown previously, the a-C:H:SiO_x_ coating deposition provides the surface with negative Coulomb potential. The higher coating thickness increases the amplitude of the negative surface potential [44], which, in turn, positively affects the in vitro reduction of pro-inflammatory cytokine and chemokine secretion by leukocytes [23]. With a view to the optimum combination of medical, biological, and mechanical properties, the a-C:H:SiO_x_ coating thickness is selected as 1.5 µm in these experiments.

### 2.2. Cytotoxicity Trials

As can be seen in Figure 5, EA.hy926 cells with the substrate extracts used at 0, 1:1, and 1:4 dilutions show the viability of 103 to 104% on T1 substrates, and 102 to 106% on T2 substrates after a 24 h incubation. The viability of EA.hy926 cells is similar to that on T1 substrates (85 ± 1.5%) and T2 substrates (84 ± 0.9%) after 72 h incubation with undiluted extracts. Moreover, for T1 and T2 substrate extracts collected at dilutions 1:1 and 1:4, cell viability is equal and ranges within 95–101% (Figure 5a,b). Therefore, some side effects could be caused in both cases by an extraction of alloying components of Ti-6Al-4V substrates.

In accordance with ISO 10993-5, the viability of a cell culture coming into contact with extracts or test samples themselves must be over 70%. Thus, the extracts of a-C:H:SiO_x_ coating exhibit no cytotoxicity against EA.hy926 cells for 72 h observation. These data are consistent with those obtained in [45,46], in which a-C:H coatings were proven to be nontoxic to human cells.

### 2.3. Cell Spreading and Distribution over Test Substrates

The number of adherent endothelial cells after the first day is similar for T1 (uncoated) and T2 (coated) substrates. After 3 days, the number of adherent cells on T2 substrates is significantly lower than that of adherent cells on T1 substrates, which continued to be the case over 7 and 14 days of observation (*p* < 0.02). For T1 substrates, the number of endothelial cells significantly decreases after 7 days (*p* = 0.01), compared to 1 day, and then increases after 14 days (*p* < 0.05). For T2 substrates, the number of adherent cells decreases after 3, 7, and 14 days (*p* = 0.01). This is shown in Figure 6. The endothelial cell distribution is also presented in Figure 7. According to Figure 6, during a period of 1 to 14 days, in vitro endothelization of the a-C:H:SiO_x_ coating on T2 substrates decreases exponentially (*y* = 2299·e^−0.971x^), with a high determination coefficient (*R*^2^ = 0.98). 

Staining of actin filaments showed that cells on the substrate surface have a flat polygonal shape with reticulated actin microfilaments. Cells more evenly distribute over the T1 substrate surface. As can be seen in Figure 7, in some endothelial cells, only rounded nuclei are observed on the T2 substrate surface, while the network of actin fibers and stress fibrils is not pronounced. Thus, the EA.hy926 cell adhesion to the T2 substrate surface is lower than the adhesion to the T1 substrate surface.

After the implantation of a medical device, its surface starts to interact with blood, and protein adsorption and cell adhesion occur. The protein adsorption and activity can be affected by physicochemical properties of the implant surface, such as free surface energy, surface wettability, electrostatic charge, and chemical composition. In [23], it was shown that the formation of the a-C:H:SiO_x_ coating on Ti-6Al-4V substrates is accompanied by an increase in the free surface energy. According to [47,48], its polar component grows with increasing Si content in the coating. In the obtained a-C:H:SiO_x_ coating with ~11 at.% Si, the free surface energy is ~30 mJ/m^2^, while the polar component is ~50% [23]. As for the Ti-6Al-4V alloy, the main contribution to the free surface energy comprises the dispersion component (~99%) [49]. Coulomb interactions play an important part in cell adhesion to the implant surface, because the coating and cell membranes are charged. As reported in [43,50,51], hemocytes and culture membranes are negatively charged, and thus are repelled from the negatively charged surface of implants. In this work, T2 substrates have a negative (−100 mV) electrostatic potential, which prevents EA.hy926 cells adhesion, as opposed to T1 substrates, which have a positive (18 mV) electrostatic potential. Due to the negative electrostatic potential of the a-C:H:SiO_x_ coating, the platelet adhesion to T2 substrates is low [22,32].

### 2.4. Optical Density of Endothelial Cell Colonization of Test Samples

The EA.hy926 cell colonization on T1 substrates after 1, 3, 5, and 7 days was higher than that on T2 substrates (*p* = 0.0004). The mass of live endothelial cells on T1 substrates did not change significantly throughout the observation period. MTT staining of endothelial cells on T2 substrates decreased after 3, 5, and 7 days compared to that after 1 day of cultivation (*p* = 0.008). The lowest density of endothelial cells on T2 substrates was 0.404 ± 0.005 after 7 days. This dependence is illustrated in Figure 8.

It is rather difficult to interpret the MTT colorimetric assay. The optical density not only helps to measure the cell viability, proliferation, and metabolic activity, but also a combination of many factors, including the MTT absorption rate and formazan extrusion [52]. Based on the cell distribution maps shown in Figure 7 for the T2 substrate, the optical density in Figure 8 is used to carefully determine changes in the cell colonies on the artificial a-C:H:SiO_x_-coated surface. Cell colonies on the surface imply an integral change in the cell mass resulting from metabolic activity, adhesion, migration, proliferation, differentiation, and death of the cell culture. Previous studies [30,53] indicate that silicon introduction in the DLC coating increases the content *sp*^3^-hybridized carbon atoms, which is accompanied by the intensification of the cell proliferation process on the coating surface. In this work, the decrease in the endothelial cell mass on the a-C:H:SiO_x_ coating was already recorded after one day of cultivation. This is likely due to the decrease in cell adhesion and proliferation after one day of cultivation. Moreover, in vivo investigations demonstrate a slight propensity of a-C:H:SiO_x_-coated substrates to endotheliocyte proliferative activity [31].

### 2.5. Endothelial Cell Death on Substrates

According to Figure 9a, the survival of endothelial cells on T1 and T2 substrates is satisfactory and varies between 75 and 90 % after 7 days of observation. This is higher than the 70% recommended by ISO 10993-5. Nevertheless, necrotic cells are found on the 3rd day of cultivation, i.e., 9 ± 7.7% and 14 ± 10.5% for T1 and T2 substrates, respectively. Surprisingly, no necrotic cells are observed on the 7th day of the in vitro experiment. One can observe a significant increase in the percentage of apoptotic cells on T1 substrates after the 7th day, compared to the 3rd day (*p* < 0.05). Moreover, the cell apoptosis on T1 substrate is higher than that on T2 substrate after 7 days (*p* = 0.04). Therefore, the initial short-time necrosis of endothelial cells contacting the artificial surface is explained by the known phenomenon of shear stress [54]. Despite this, the a-C:H:SiO_x_ coating on T2 substrate prevents cell damage with time. This is demonstrated by the number of live cells on the 7th day being higher (up to 90%) than that on the 3rd day (75%, *p* < 0.05). Figure 9b presents maps of endothelial cell distribution over T1 and T2 substrates.

Therefore, the reduction in MTT staining indices (see Figure 8) of EA.hy926 cells on a-C:H:SiO_x_-coated substrates can be determined as a decrease in the cell mass, rather than the survival of cells contacting with the artificial surface.

The cell apoptosis reduction and viability increase on the a-C:H:SiO_x_-coated substrate surface can be explained by its hydrophilicity after the coating deposition. The high content of Si–O bonds increases the polar component of the surface free energy and changes the surface wettability. The higher the Si content, the higher the wettability. According to Bociaga et al. [30], the cell viability on the coating surface is enhanced as a result of its increased hydrophilicity.

The decrease in the cell number (Figure 6 and Figure 7) and mass (Figure 8) on the a-C:H:SiO_x_ coating is most likely stipulated by their lower adhesion, rather than their viability (Figure 9).

### 2.6. Nitric Oxide Production by Endothelial Cells

The natural function of the endothelial tissue is mostly determined by its ability to synthesize NO, which is one of the main control factors [55,56,57]. Basal NO release from the endothelial tissue controls the vascular tone and counteracts the activity of vasoconstrictor agents. In addition, NO provides an antithrombotic effect and suppresses the expression of adhesive molecules, which promote inflammatory cell adhesion [58]. The use of pharmacological agents that suppress endotheliocyte proliferation over the surface of intravascular implants can also significantly suppress NO synthesis, both in early post-implantation and over long-term periods, due to the development of genetically determined cell changes [59,60].

The highest NO level is detected after a 24 h cultivation of endothelial cells on T1 substrates and plastic. It should be noted that the level of NO production by cells cultured on T2 substrates for 24 h is lower than that by those cultured on T1 substrates (*p* = 0.02). After a 72 h cell cultivation, the NO production considerably decreases on T1 and T2 substrates and on the plastic (*p* < 0.01). As can be observed in Figure 10, the NO production by endothelial cells cultured on T2 substrates is much lower than that by those cultured on plastic after 24 and 72 h (*p* = 0.01). The low level of nitric oxide on T2 substrates is associated with the reduced distribution of these samples by endothelial cells (see Figure 8). No increased cytotoxicity of T2 substrates, compared with T1 control samples, was found after 72 h (3 days) of observation (Figure 9a).

The reduced NO production confirms the results of the MTT colorimetric assay concerning the reduced colonization of EA.hy926 cells on a-C:H:SiO_x_-coated substrates, as endothelial cells including EA.hy926 produce nitric oxide molecules [61,62]. The EA.hy926 cell survival can enhance after 7 days (see Figure 9a) against the lower NO production, since high nitric oxide doses can induce death of endothelial cells, owing to pronounced oxidative stress [11].

On the other hand, NO causes vasorelaxation [63] and decreases the inflammatory cell adhesion to the vascular endothelium [56]. Therefore, further research will be needed in order to study the integral biological significance of the NO in vitro production by EA.hy926 endothelial cells adhered to the a-C:H:SiO_x_ coating.

## 3. Materials and Methods

### 3.1. Preparation of Ti-6Al-4V Substrate and Coating Deposition

Ti-6Al-4V substrates (VSMPO-AVISMA Corporation, Russia) with a diameter of 5 mm or 10 mm, and 1 mm thick, were used in this experiment. Prior to the a-C:H:SiO_x_ coating deposition, the substrates were mechanically polished on P2000 sandpaper, and then cleaned ultrasonically in isopropyl alcohol, acetone, and distilled water. In each liquid, the substrates were treated for 10 min. The a-C:H:SiO_x_ coating was synthesized in a vacuum by plasma-assisted chemical vapor deposition (PACVD) from a polyphenylmethylsiloxane (PPMS) precursor using the pulsed bipolar bias voltage of the substrate. Figure 11 shows the stages of sample preparation and a-C:H:SiO_x_ coating deposition. The process parameters are summarized in Table 1. A thorough description of the PACVD system is given in [64]. For comparison, Ti substrates were divided into two groups, namely uncoated (T1) and a-C:H:SiO_x_-coated (T2) Ti-6Al-4V substrates.

### 3.2. Structural Analysis of the a-C:H:SiO_x_ Coating

The surface morphology and elemental composition of the a-C:H:SiO_x_ coating were studied with a Quanta 200 3D (FEI Company, Hillsboro, Oregon, USA) scanning electron microscope (SEM), coupled with an energy dispersive X-ray analyzer.

The coating microstructure was investigated on the JEM-2100 (JEOL Ltd., Tokyo, Japan) high-resolution transmission electron microscope (TEM) in bright and dark fields, using thin film technology for micro/nano systems in order to obtain X-ray diffraction (XRD) imaging. The accelerating voltage was 200 kV. The preparation of cross-sectional TEM images of thin films was described in [65], which implied film thinning to ~200 µm on an EM-09100IS Ion Slicer (JEOL Ltd., Tokyo, Japan), preliminary cut normal to the film plane. That allowed study of the microstructure inside the cross-section at a different distance from the surface and XRD pattern measurement at a certain depth step.

The analysis of the atomic short-range order of amorphous structures in the synthesized coating was performed via the function recovery of atomic radial distribution by experimental nano-beam diffraction (NBD) according to [66]. The size of the NBD area was 25 nm. The quantitative analysis of coordination shell radii and coordination numbers was based on calibration XRD patterns of the gold replica and the chemical composition of XRD-analyzed regions, and conducted on an INCA (Oxford Instruments, Abingdon, GB) energy dispersive X-ray (EDX) system, integrated in TEM. The EDX/TEM errors included ±3 at.% Si, ±4 at.% O, and ±5 at.% C. Measurement error of the atomic spacing was not over 0.003 nm, whereas for coordinate numbers, there was an accuracy was ±0.6. The radial distribution function (RDF) of atoms was obtained using Origin 9 software.

### 3.3. Preparation of Extracts

For biological studies, all 60 substrates were sterilized in a steam sterilizer (BMG Labtech, Ortenberg, Germany) at 121 °C and 0.5 atm. for 45 min. The distribution of test samples for in vitro investigation is presented in Table 2.

In order to study the indirect cytotoxicity of substrates, their extracts were prepared in accordance with ISO 10993-5 and ISO 10993-12. The uncoated Ti-6Al-4V substrates (T1) and a-C:H:SiO_x_-coated substrates (T2) with a diameter of 5 mm (6 samples per each group) were incubated in 1.7 mL of DMEM/F-12 complete growth medium (Gibco, Carlsbad, CA, USA) supplemented with 10% FCS (Hyclone, Logan, UT, USA), 1% GlutaMAX (Gibco, Carlsbad, CA, USA), and 40 µg·mL^−1^ gentamicin (Dalkhimpharm, Khabarovsk, Russia), at a surface-area-to-volume ratio of 1.25 cm^−1^. Incubation was done at 37 °C over 72 h. By the endpoint, the substrates were removed from the medium, and extracts were collected at 0, 1:1, and 1:4 dilutions in order to determine cytotoxicity with an endothelial cell culture.

### 3.4. Cell Culture

The EA.hy926 cell line was kindly provided by Dr. Edgell from Carolina University (Winston-Salem, NC, USA). The human hybrid EA.hy926 cells were cultured in the Gibco™ DMEM/F-12 supplemented with 10% FCS, 1% GlutaMAX, and 40 µg·mL^−1^ gentamicin, with a replacement of the culture medium every 3 or 4 days. After the formation of the confluent monolayer during passaging, adherent cells were detached using a 0.25% trypsin (BioloT, Saint Petersburg, Russia) and twice washed in a phosphate buffer solution (PBS) (BioloT, Saint Petersburg, Russia).

### 3.5. Cytotoxicity Analysis

EA.hy926 cells were seeded in 96-well plates, i.e., 1 × 10^4^ cells per well for 24 h to allow attachment. Subsequently, the nutrient medium was removed, and 100 µL of extract alone, or its dilutions of 0, 1:1, 1:4, were added to each well. The calorimetric analysis was based on the intracellular 3-(4,5-dimethylthiazol-2-yl)-2,5-diphenyl-2H-tetrazolium bromide (MTT) assay (Sigma-Aldrich, Darmstadt, Germany) in order to evaluate cell viability after 24 and 72 h, in accordance with ISO 10993-5. By the end of the incubation period, the culture medium was removed from wells, and the fresh Gibco™ DMEM/F-12 and MTT dye (5 mg·mL^−1^) solution were added in the amounts of 90 and 10 µL, respectively. After 4 h, the dye was removed, and formazan crystals were dissolved by adding 100 μL dimethyl sulfoxide (PanReac AppliChem, Darmstadt, Germany). The absorbance of dissolved formazan crystals was measured at 492 nm using a Stat Fax-2100 Plate Reader (Awareness Technology Inc., Palm City, FL, USA). Cell viability was calculated using the equation (experimental group *A*/control group *A*) × 100%, where *A* is the optical density. Intact cells cultivated in the Gibco™ DMEM/F-12, were used as reference cells.

### 3.6. Cell Distribution over the Surface

A Vybrant™ CFDA SE (carboxy fluorescein diacetate succinimidyl ester) Cell Tracer Kit (Thermo Fisher Science, Waltham, MA, USA) was used in a separate experiment to detect long-term non-toxic fluorescence of cells adhered to the surface of T1 and T2 substrates (*n* = 6 in each group). Before cell seeding onto substrates, they were pre-incubated with the CFDA SE for 30 min according to the manufacturer’s instructions. EA.hy926 cells were applied to the substrate surface (12 × 10^4^ cells in 60 μL for a substrate) and cultivated for 14 days. The medium was changed every 2 or 3 days. On the 1st, 3rd, 7th, and 14th day, cells were counted per at least 5 fields of view using an Axio Observer Fluorescent Microscope (Zeiss, Oberkochen, Germany). The results were expressed via the adherent cell number per 1 mm^2^ of the test surface.

Vybrant™ CFDA SE dye binds covalently to all free amines on the surface and inside of cells according to manufacturer staining protocol. Some cells may be refractory to Vybrant dye uptake irrespective of concentration or duration of treatment [67]. Therefore, cell staining with other dyes is needed (see below).

In addition, the cell actin cytoskeleton was stained with phalloidin and 4′,6-diamidino-2-phenylindole (DAPI) (Abcam, Cambridge, MA, USA) after 7 days of cultivation. According to the manufacturer’s instructions, phalloidin conjugated to Alexa Fluor™ 488 (Thermo Fisher Scientific, Waltham, MA, USA) was incubated at 1:200 dilution in the PBS for 1 h. The surface of T1 and T2 substrates was then imaged using the Axio Observer.

### 3.7. Cell Survival Estimation

Titanium substrates (*n* = 6 in each group) were placed in a 24-well plate. Cells were applied (12 × 10^4^ cells in 60 µL per a substrate) and cultured for 1, 3, 5, and 7 days. The nutrient medium was changed every 2 or 3 days. The MTT colorimetric assay was used to determine the cell survival at the cell/substrate interface, as described in paragraph 2.4. The color intensity of dissolved formazan crystals was proportional to the number of live cells.

### 3.8. Cell Death Estimation

Substrates (*n* = 6 in each group) were placed in a 24-well plate (one to a well), and endothelial cells were applied (12 × 10^4^ cells in 60 µL per substrate) and cultured for 3 and 7 days. The medium was changed every 2 or 3 days. Endothelial cell apoptosis was identified by acridine orange (DIA M, Moscow, Russia) staining (100 μg·mL^−1^) and ethidium bromide (Medigen, Novosibirsk, Russia) staining (100 μg·mL^−1^). The substrates were then analyzed on the Axio Observer. At least 500 cells on the substrate were counted for the assay. As a result, the number of live, necrotic, or apoptotic cells was calculated, stained respectively green, red, or yellow [68], and their percentages were included.

### 3.9. Nitric Oxide Assay

Nitric oxide (NO) production was analyzed by measuring the nitrite level as a stable NO product using the Griess reagent (Sigma-Aldrich, Darmstadt, Germany) in accordance with the manufacturer’s instructions. EA.hy926 cells were seeded on the surface of T1 and T2 substrates (*n* = 6 in each group) in a 24-well plate (12 × 10^4^ cells per sample). After a 24 or 72 h cultivation, supernatants were harvested, and a mixture of 50 μL supernatant and 50 μL Griess reagent was added to the 96-well plate. The cell absorbance at 492 nm was identified by the Stat Fax-2100. The nitrite concentration was measured by a standard calibration curve.

### 3.10. Statistical Analysis

Statistical hypothesis testing was applied to the obtained data using the Statistica 10.0 software package (TIBCO Software Inc., Palo Alto, CA, USA) for Windows. The obtained data were presented as mean ± SD (standard deviation). The Shapiro-Wilk test was used to control the normal distribution of all variables. Since the obtained data did not show a normal distribution, a nonparametric test was appropriate, namely the Mann–Whitney *U* test. The *R* regression analysis was also used for estimating relationships between variables. The statistical significance between the groups was established at *p* < 0.05.

## 4. Conclusions

TEM investigations showed that the synthesized a-C:H:SiO_x_ coating possessed an amorphous structure and consisted of Si–C, Si–O, Si–Si, and C–C bonds. The contribution of each bond determined the properties of the obtained coating, since their nature was different (polar/nonpolar), as were their electronegativity and other parameters. These key factors affected the surface wettability, Coulomb potential, and interaction between endothelial cells and the a-C:H:SiO_x_ coating.

Based on the results, it can be concluded that the a-C:H:SiO_x_ coating had no clear cytotoxicity relative to EA.hy926 cells. In the course of in vitro studies, it was shown that the adhesion of EA.hy926 cells to a-C:H:SiO_x_-coated substrates was significantly lower than their adhesion to uncoated substrates. When staining the network of actin fibers and stress fibrils of endothelial cells, the contact area between endothelial cells and the coating surface was clearly minimized, as well as weakly expressed network of actin fibers and stress fibrils in cells occurred. The low level of NO production on the coated substrates was associated with a much lower colonization of a-C:H:SiO_x_ coating by endothelial cells.

With regard to previous research into hypothrombogenic properties, the a-C:H:SiO_x_ coating can be advantageously used to modify the surface of endovascular implants, such as heart pumps and artificial heart valves. The a-C:H:SiO_x_ coating application can reduce the hazard of hyperproliferation and the migration of cellular elements, thereby reducing the risk of implant restenosis without increasing the risk of thrombotic complications.

## Figures and Tables

**Figure 1 ijms-24-06675-f001:**
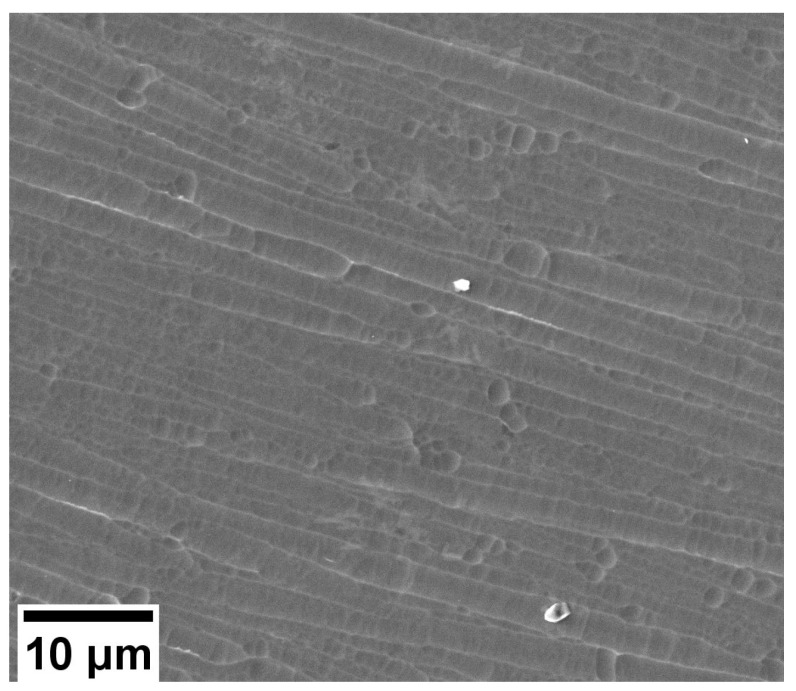
SEM image of the a-C:H:SiO_x_ coating surface on Ti-6Al-4V substrate.

**Figure 2 ijms-24-06675-f002:**
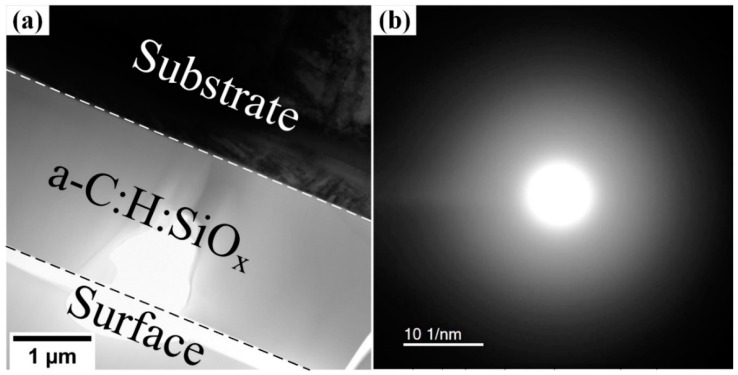
Cross-sectional bright-field TEM image of a-C:H:SiO_x_ coating (**a**), NBD at the coating center (**b**).

**Figure 3 ijms-24-06675-f003:**
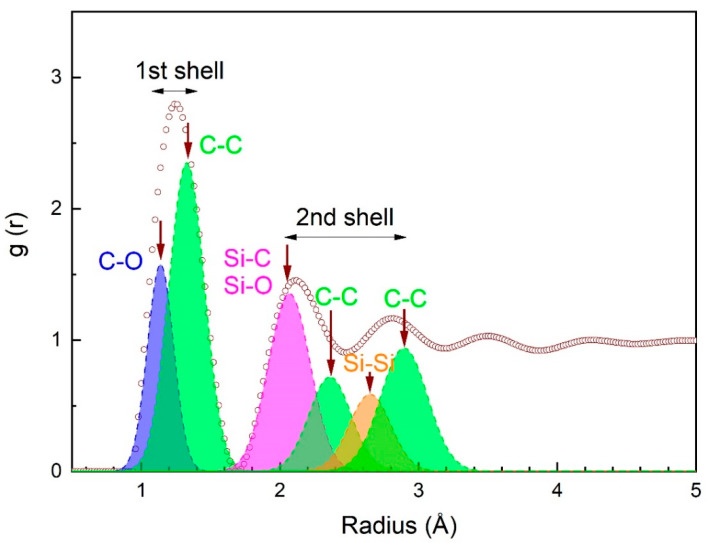
RDF of atoms in a-C:H:SiO_x_ coating and peak resolution into pair atomic spacings using the Gaussian function. Five areas were examined for statistics. X axis indicates the radial distance (r) (given in angstrom). Y axis indicates radial distribution function g(r) (given in arbitrary units).

**Figure 4 ijms-24-06675-f004:**
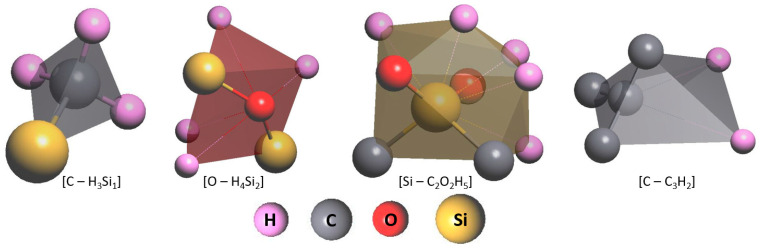
The model (pattern) of the short-range order atomic structure of a-C:H:SiO_x_ coating.

**Figure 5 ijms-24-06675-f005:**
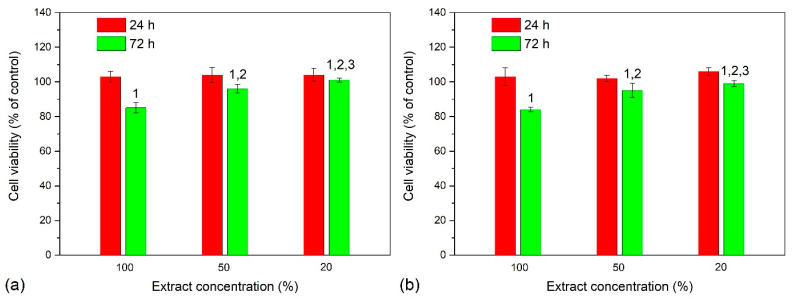
Viability of EA.hy926 cells after 24 and 72 h incubation with extracts: (**a**)—T1 substrates, (**b**)—T2 substrates. Data are presented as mean ± SD. 1—*p* < 0.05 compared to 24 h, 2—*p* < 0.02 compared to 100% extract after 72 h, 3—*p* < 0.05 compared to 50% extract after 72 h. Each group consists of six substrates for testing.

**Figure 6 ijms-24-06675-f006:**
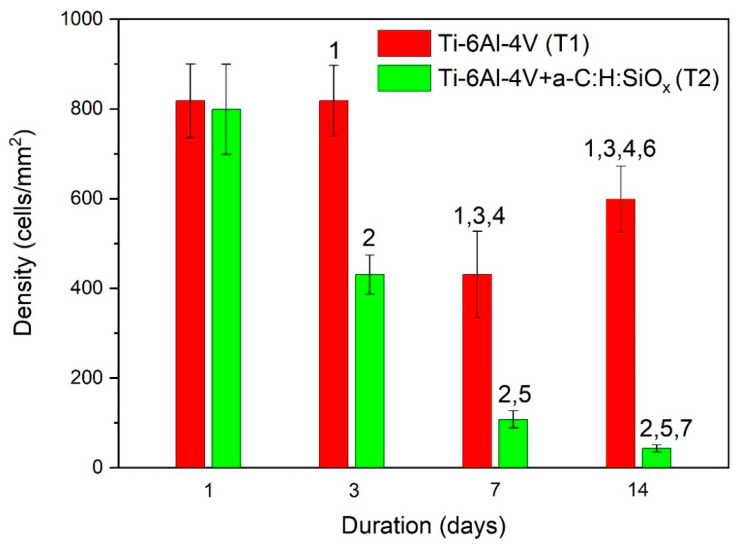
Density distribution of EA.hy926 cells depending on days of substrate cultivation: 1—T1 vs. T2 after 3, 7, and 14 days (*p* < 0.02), 2—T1 vs. T2 after 1 day (*p* = 0.01), 3—T2 vs. T1 after 1 day (*p* = 0.01), 4—T2 vs. T1 substrate after 3 days (*p* < 0.02), 5—T1 vs. T2 substrate after 3 days (*p* = 0.01), 6—T2 vs. T1 substrate after 7 days (*p* < 0.05), 7—T1 vs. T2 substrate after 7 days (*p* = 0.01). Data are presented as mean ± SD. Each group consists of six substrates for testing.

**Figure 7 ijms-24-06675-f007:**
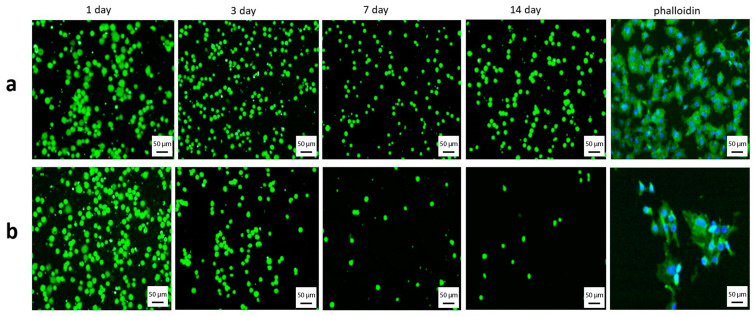
EA.hy926 cell distribution maps on substrates: (**a**)—T1 surface, (**b**)—T2 surface after culturing of 1 to 14 days. Scale bar: 50 µm. Adherent cells, actin filaments, and nuclei are stained with Vybrant™ CFDA SE Cell Tracer Kit (green), phalloidin (green), and DAPI (blue), respectively.

**Figure 8 ijms-24-06675-f008:**
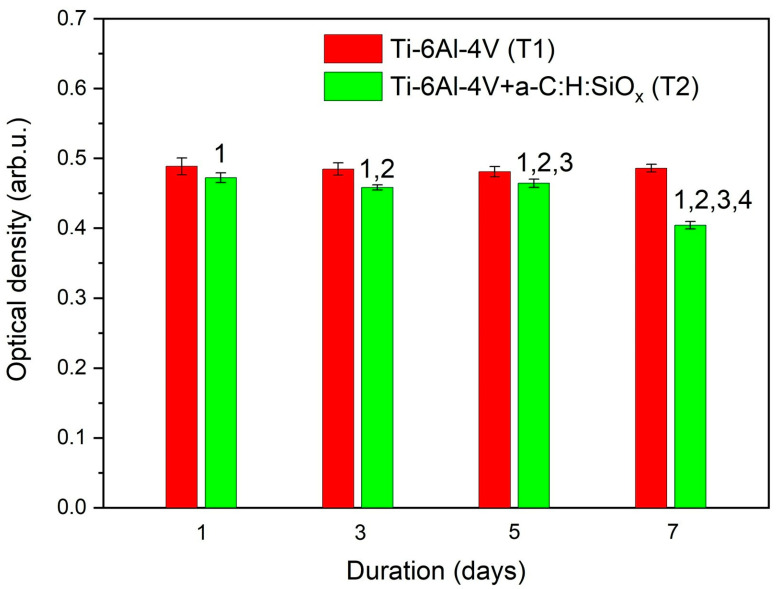
Optical density distribution of EA.hy926 cells depending on days of substrate cultivation: 1—T2 vs. T1 after 1, 3, 5, and 7 days (*p* = 0.0004), 2—T1 vs. T2 after 1 day (*p* = 0.04), 3—T1 vs. T2 after 3 days (*p* < 0.04), 4—T1 vs. T2 after 5 days (*p* = 0.008). MTT staining is presented as mean ± SD. Six measurements were performed in each group. *Y* axis indicates dissolved formazan crystals.

**Figure 9 ijms-24-06675-f009:**
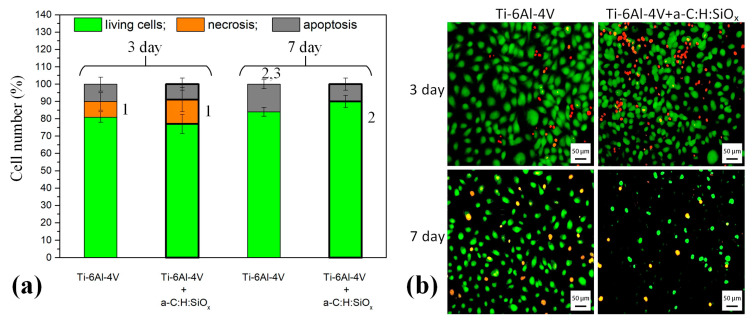
Block diagram (**a**) of EA.hy926 cell death on Ti substrates: 1—necrotic cells vs. those after 7 days (*p* = 0.01), 2—live cells vs. those on T1–T2 substrates after 3 days (*p* < 0.05), 3—live cells vs. those on T2 substrate after 7 days (*p* = 0.04). Data are presented as mean ± SD. Six measurements were performed in each group. Cell distribution maps (**b**) on T1 and T2 substrates. Staining: acridine orange and ethidium bromide. Scale bar: 50 µm. Magnification: 200×.

**Figure 10 ijms-24-06675-f010:**
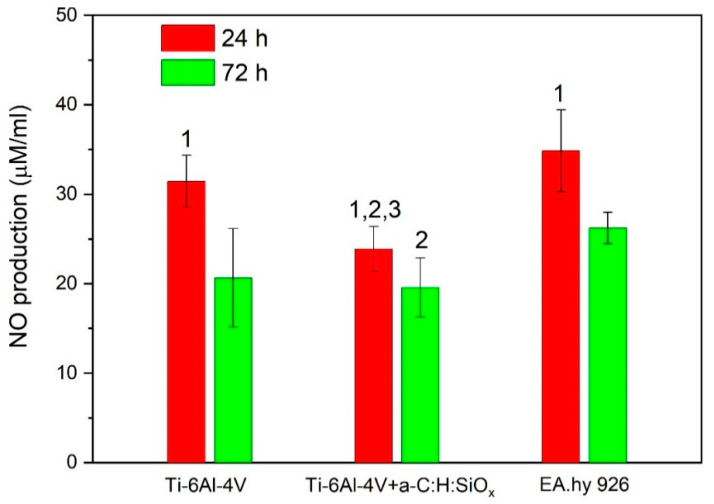
NO production by EA.hy926 cells cultured on substrates: 1—24 h vs. 72 h (*p* < 0.01), 2—T2 vs. EA.hy926 cells on culture plastic (*p* = 0.01), 3—T2 vs. T1 (*p* = 0.02). Data are presented as mean ± SD. Six measurements were performed in each group. Control-EA.hy926 cells cultured on plastic.

**Figure 11 ijms-24-06675-f011:**
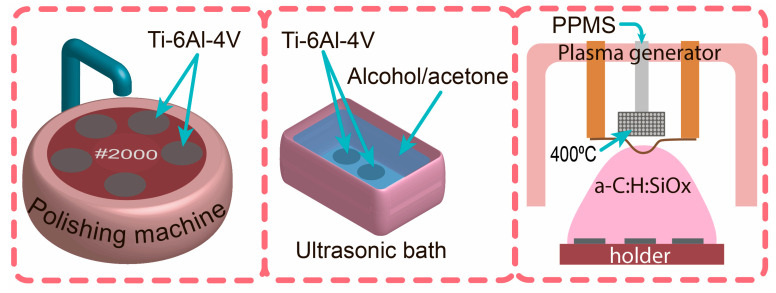
Stages of sample preparation and a-C:H:SiO_x_ coating deposition.

**Table 1 ijms-24-06675-t001:** PACVD process parameters.

Parameters	Stage I (Ion-Beam Cleaning)	Stage II (Coating Deposition)
Ar pressure, Pa	0.3 ± 0.01	0.1 ± 0.01
Discharge current, A	7 ± 0.5	5.2 ± 0.3
Filament current, A	50 ± 5	50 ± 5
Bias voltage, V	1000 ± 50	300 ± 20
Ar flow rate, sccm	230 ± 10	66 ± 5
Precursor flow rate, ×10^−3^ sccm	-	17 ± 1
Substrate temperature, °C	20 ÷ 200	170 ± 5
Duration, min	15	120

**Table 2 ijms-24-06675-t002:** Sample distribution for in vitro biomedical examination.

Substrate Material	Number of Test Samples in Each Group				
	Extract Cytotoxicity (Discs, 5 mm in Diameter)	Cell Spreading and Distribution	Optical Density of Endothelial Cell Colonization	Endothelial Cell Death (Apoptosis, Necrosis)	Nitric Oxide Production by Endothelial Cells
Ti–6Al–4V alloy (T1)	6	6	6	6	6
Ti–6Al–4V alloy + a-C:H:SiO_x_ (T2)	6	6	6	6	6

Series of experiment with separate cell passages are colored differentially.

## Data Availability

The data presented in this study are available on request from the corresponding author.

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
