# Peer review of "Endothelial Cell Behavior and Nitric Oxide Production on a-C:H:SiOx-Coated Ti-6Al-4V Substrate"

_ijms, 2023, doi:10.3390/ijms24076675_

Round 1

Reviewer 1 Report

The surface of Ti-6Al-4V was modified by a-C:H:SiOx 16 via coating deposition. The coating structure was investigated by several instrumental methods and in vitro endothelization. The results indicate that the a-C:H:SiOx 25 coating can reduce the risk of endothelial cell hyperproliferation on implants and medical devices, which are important for the development of biomedicne. This study is interesting and the findings are fine and useful.

A minor revision is recommended:

The intruduction is too long and should be shortened.

It is necessary to add a scheme to illustrate the procedure to prepare Ti-6Al-4V substrate and coating deposition

Aviod to use the expression of we.... throughout the manuscript.

Pay attention to the tense of each sentence.

The results are good and have been well discussed.

Author Response

Detailed Response to Reviewers

Journal: International Journal of Molecular Sciences

Title: ijms-2289916 Endothelial cell behavior and nitric oxide production on a-C:H:SiOx-coated Ti-6Al-4V substrate

Dear Respected Editor,

We would like to convey our sincerest gratitude to you and the respected reviewers for their valuable comments to improve the manuscript. The corrections are highlighted in green color in this revised version of the manuscript. Here, we also have added below the answers next to the queries raised by the reviewers.

Responses to Reviewer #1’s comments:

The surface of Ti-6Al-4V was modified by a-C:H:SiOx 16 via coating deposition. The coating structure was investigated by several instrumental methods and in vitro endothelization. The results indicate that the a-C:H:SiOx 25 coating can reduce the risk of endothelial cell hyperproliferation on implants and medical devices, which are important for the development of biomedicne. This study is interesting and the findings are fine and useful.

Question 1. The intruduction is too long and should be shortened.

Response: We strongly respect the opinion of the Reviewer. However, the topic is interdisciplinary and requires initial explanations of relevance for both biologists and materials scientists.

Question 2. It is necessary to add a scheme to illustrate the procedure to prepare Ti-6Al-4V substrate and coating deposition

Response: The required figure is added (Fig. 11).

Question 3. Aviod to use the expression of “we....” throughout the manuscript.

Response: The expression of “we....” is corrected throughout the manuscript.

Question 4. Pay attention to the tense of each sentence.

Response: The final version of the manuscript will be checked by English language specialists from MDPI.

Question 5. The results are good and have been well discussed.

Response: We thank dear Reviewer for positive opinion on our paper. Blue marker colored our minor revisions and additions in the text.

Thank you.

Yours sincerely,

Prof. Igor A. Khlusov, Maria A. Surovtseva, Alexander S. Grenadyorov and Andrey A. Solovyev.

Reviewer 2 Report

The authors tested cell adhesion and cytotoxicity on an previously reported a-C:H:SiOx coating on Ti-6Al-4V substrate. However, some of the data are not clearly presented and/or contradictory.

A material ration of 85 at.% C, 11 at.% Si, and ~4 at.% O of the coating was described to be optimal. The author did not clarify whether this optimisation is part of this study or a result from previous studies. If yes, more data and explanation of the optimisation should be presented. If no, references are needed.

The authors then tested the biocompatibility of the material using the EA.hy926 cell line. Figure 6 and 7 presented clear data of a 10-fold reduction of cells on the material over a period of 14 days. Very few cells remained on the a-C:H:SiOx-coated surface at day 7 and 14. The initial high adhesion followed by a rapid reduction in cell number certainly raises concerns of some sort of cytotoxic effects. In Figure 5, the conditioned medium has shown some cytotoxic effects at 100%. Co-culture of insert well with the coated/uncated material may be a more robust method to exam cytotoxicity and to monitor for longer period of culture (7-14 days).

Figure 9 (b) presented the EA.hy926 cells are nearly confluent at day 7 on the a-C:H:SiOx-coated surface. This is directly contradictory to figure 6 and 7. Figure 10 suggests cells on a-C:H:SiOx-coated surface produced similar amount of NO versus control surface at day 3. However, the cell density is only about 50% on a-C:H:SiOx-coated versus control as showed in Figure 7. This suggests that NO production was doubled by each cell on a-C:H:SiOx-coated surface versus control, which is confusing and unexplained.

Collectively, the data presented in this manuscript is contradictory and insufficient to support the main conclusion that the surface is non-toxic and non-adherent.  

Minor points:

Line 74-75: The sentence is not clear to me. Please revise.

Line 344, table 1. Typo for ‘Substrate temperature, °C    20÷200  ?

Author Response

Detailed Response to Reviewers

Journal: International Journal of Molecular Sciences

Title: ijms-2289916 Endothelial cell behavior and nitric oxide production on a-C:H:SiOx-coated Ti-6Al-4V substrate

Dear Respected Editor,

We would like to convey our sincerest gratitude to you and the respected reviewers for their valuable comments to improve the manuscript. The corrections are highlighted in green color in this revised version of the manuscript. Here, we also have added below the answers next to the queries raised by the reviewers.

Reviewer #2:

The authors tested cell adhesion and cytotoxicity on an previously reported a-C:H:SiOx coating on Ti-6Al-4V substrate. However, some of the data are not clearly presented and/or contradictory.

Question 1. A material ration of 85 at.% C, 11 at.% Si, and ~4 at.% O of the coating was described to be optimal. The author did not clarify whether this optimisation is part of this study or a result from previous studies. If yes, more data and explanation of the optimisation should be presented. If no, references are needed.

Response: Information about the influence of the deposition conditions and optimum modes is given in Section 2.1 together with references to our previous research (see the first paragraph). You can read there that optimization of the deposition mode was based on high mechanical, tribological, and anticorrosion properties of the Ti-6Al-4V/a-C:H:SiOx system.

Question 2. The authors then tested the biocompatibility of the material using the EA.hy926 cell line. Figure 6 and 7 presented clear data of a 10-fold reduction of cells on the material over a period of 14 days. Very few cells remained on the a-C:H:SiOx-coated surface at day 7 and 14. The initial high adhesion followed by a rapid reduction in cell number certainly raises concerns of some sort of cytotoxic effects. In Figure 5, the conditioned medium has shown some cytotoxic effects at 100%. Co-culture of insert well with the coated/uncated material may be a more robust method to exam cytotoxicity and to monitor for longer period of culture (7-14 days).

Response: At first view, there may indeed to be contradictions in the results in different sections of the initial text of the manuscript. We have seriously revised our article as follows:

2.1. We have added Table 2 to section 3 with the designation of 4 series of study using cells from different passages. This was due to the need to conduct tests that cannot be performed simultaneously in single experiment. At the same time, it is incorrect to compare the results of different reactions of the same cell line in different passages.

2.2. We have changed the description of section 2.2 to increase the reader's understanding that the coating extracts do not have 72-hour cytotoxicity compared to uncoated substrate. In this aspect, the negative effect of 100% extract is most likely due to the release of alloying components from Ti-6Al-4V alloy (V, Al, Fe, etc.), which is approved for medicine and has long been used in surgical practice. Consequently, consumer properties significantly prevail over the known undesirable consequences of using Ti-6Al-4V alloy in biomedicine.

2.3. We have changed the description of some other sections, which we will answer in question #3.

As for Figures 6-7, the Vybrant™ CFDA SE Cell Tracer Kit used in section 2.3 only tells about the number of adherent cells on the surface of the samples. The number of adherent cells depends on their overall viability, as well as on the activity of the cytoskeleton. In a separate study with Vybrant™ dye, the number of endothelial cells was greatly reduced on the coating. We added to section 3.6 the fact that the results of staining with Vybrant has their own artifacts, which should be taken into account; therefore other cell staining methods should be used.

Indeed, in MTT staining (Fig. 8), the cell colonization of a-C:H:SiOx coating already decreased slightly; when we are determining cell death (Fig. 9) using conventional dyes (acridine orange and ethidium bromide), there are no differences between T1 and T2 samples at all.

We do not hide from the text of the article that there is a certain negative effect of a-C:H:SiOx coating on the adhesion of endothelial cells. However, it is associated more with decreased cytoskeletal activity (Fig. 7 with phalloidin) than with phenomena of specimen’s cytotoxicity. Our opinion based on the results is presented in lines 299-301 (pdf-file of the article) as follows: The decrease in the cell number (Figs. 6, 7) and mass (Fig. 8) on the a-C:H:SiOx coating, 299 is most likely stipulated by their lower adhesion and proliferation, rather than their viability (Fig. 9).

We removed the term “proliferation” from this sentence, because it is not properly studied in our article.

Question 3. Figure 9 (b) presented the EA.hy926 cells are nearly confluent at day 7 on the a-C:H:SiOx-coated surface. This is directly contradictory to figure 6 and 7. Figure 10 suggests cells on a-C:H:SiOx-coated surface produced similar amount of NO versus control surface at day 3. However, the cell density is only about 50% on a-C:H:SiOx-coated versus control as showed in Figure 7. This suggests that NO production was doubled by each cell on a-C:H:SiOx-coated surface versus control, which is confusing and unexplained.

Response: Question 3 is a continuation of question 2. Therefore, we emphasized again that Figures 6-7 and 10 refer to different series of studies according to Table 2 that we inserted into the section 3.3. The functional activity of the same cell cultures often changes in different passages, which depends on many factors and is not always explained. Hence, only the results in Figures 9 and 10 can be accurately compared, since this is one cell passage.

Accordingly, T2 coated samples do not enhance the cytotoxicity of Ti-6Al-4V substrates (Fig. 9), so the decrease in NO production may be associated with weakened cell adhesion to a-C:H:SiOx coating. NO production is reduced by the 3rd day also in the control culture on plastic (Fig. 10); both coated and uncoated samples only slightly enhance this effect. Therefore, the general biological behavior of EA.hy926 cells in in vitro culture cannot be excluded. It may depend on not only the viability and number of cells, but for example, on the activity of nitric oxide synthase (eNOS). Consequently, at the end of section 2.6 we noted that further research into the issue of NO production is needed (lines 330-332 of the pdf-file of the manuscript).

Question 4. Collectively, the data presented in this manuscript is contradictory and insufficient to support the main conclusion that the surface is non-toxic and non-adherent.

Response: We hope that our corrections in the text will allow dear Reviewer to change his mind about the inconsistency of the results. In the revised version of our article, we have eliminated and/or explained the phantom contradictions.

Minor points:

Question 5. Line 74-75: The sentence is not clear to me. Please revise.

Response: We revised this sentence as follows: However, the complete endothelization of blood-contacting surfaces of cardiovascular devices (camera, moving parts) is not always optimum to prevent post-implantation complications.

Question 6. Line 344, table 1. Typo for ‘Substrate temperature, °C    20÷200’  ?

Response: There is no inaccuracy here. Table 1 shows the temperature at Stage 1. It means that at the initial moment, specimens possess room temperature (20оС). During ion cleaning at this stage, they are gradually heated. After 15 min, their final temperature is 200оС. No forced heating is applied during the ion cleaning. Heating occurs due to the ion impact intensified by the bias potential applied to the substrate.

Thank you.

Yours sincerely,

Prof. Igor A. Khlusov, Maria A. Surovtseva, Alexander S. Grenadyorov and Andrey A. Solovyev.

Round 2

Reviewer 2 Report

My original comment was not addressed.

The picture of cell adhesion on the coated surface in Figure 9b (day 7) is drastically different to the Figure 6 and figure 7b (day 7). 

Figure 9b suggests very good cell adhesion/survival on the coated surface over 7 days. Fig6 and 7 suggest a drastic loss of cells on the coated surface. These data are contradictory. 

Author Response

Detailed Response to Reviewers

Journal: International Journal of Molecular Sciences

Title: ijms-2289916 Endothelial cell behavior and nitric oxide production on a-C:H:SiOx-coated Ti-6Al-4V substrate

Dear Respected Editor,

We would like to convey our sincerest gratitude to you and the respected reviewers for their valuable comments to improve the manuscript. The corrections are highlighted in green color in this revised version of the manuscript. Here, we also have added below the answers next to the queries raised by the reviewers.

Reviewer #2:

My original comment was not addressed.

Response:  We thank again dear Reviewer for discussion of our article. 

Question 1. The picture of cell adhesion on the coated surface in Figure 9b (day 7) is drastically different to the Figure 6 and figure 7b (day 7).

Response:  Our comment and corrections on the visual differences between Figures 7b and 9b are presented below in the answer to question 2.

Regarding the differences in Figs 6 and 9a, we have presented our green comments in section 3.6. of the articles and in our response in round 1 as follows: “Vybrant™ CFDA SE dye binds covalently to all free amines on the surface and inside of cells according to manufacturer staining protocol. Some cells may be refractory to Vybrant dye uptake irrespective of concentration or duration of treatment [68]. Therefore, cell staining with other dyes is needed (see below)”.

Herein, the Vybrant stains cells selectively; therefore, these results cannot be compared closely with other dyes (MTT, acridine orange, and ethidium bromide), which are also used for different purposes.

Besides, Fig.6 presents cell number per surface area (cells/mm2); Fig.9a shows the percent (%) of calculated died and live cells independent of number of adherent cells. These are different indicators that cannot be compared closely.

Question 2. Figure 9b suggests very good cell adhesion/survival on the coated surface over 7 days. Fig6 and 7 suggest a drastic loss of cells on the coated surface. These data are contradictory.

Response: Micrographs of cells are usually intended for visual subjective confirmation of digital and graphical results. The number of cells can vary significantly in different fields of view. We modified Fig. 9b (7 days) to present a fields of view with lesser amount of adherent cells to overcome the visual differences in Figs 7b and 9b. Now, there are no visual contradictions between the new version of Fig. 9b and Fig. 7b.

We believe that with the new Fig.9b and our comments above, we have explained to dear Reviewer the absence of contradictories in our data.

Thank you.

Yours sincerely,

Prof. Igor A. Khlusov, Maria A. Surovtseva, and Alexander S. Grenadyorov.

Round 3

Reviewer 2 Report

In the two rounds of responses, the author attributed the contradictory data to the difference of cell passage and problems of the staining methods (Vybrant™ CFDA SE dye ). If this was true, it only suggests the reproducibility of the study is very poor and the choice of experimental methods was inappropriate. 

The authors claimed they modified Fig 9b. But I don't think I can identify any changes in the revised manuscript v2.

The scale bar in Fig 7 is clearly incorrect in some panels.

Author Response

Detailed Response to Reviewers

Journal: International Journal of Molecular Sciences

Title: ijms-2289916 Endothelial cell behavior and nitric oxide production on a-C:H:SiOx-coated Ti-6Al-4V substrate

Dear Respected Editor,

We would like to convey our sincerest gratitude to you and the respected reviewers for their valuable comments to improve the manuscript. The corrections are highlighted in green color in this revised version of the manuscript. Here, we also have added below the answers next to the queries raised by the reviewers.

Comments and Suggestions for Authors

Question 1. In the two rounds of responses, the author attributed the contradictory data to the difference of cell passage and problems of the staining methods (Vybrant™ CFDA SE dye ). If this was true, it only suggests the reproducibility of the study is very poor and the choice of experimental methods was inappropriate.

Response: This remark of a respected reviewer is vague, in our opinion. The reasoning is too emotional; the reviews need factual claims.Certain limitations of any dye and staining methods are not true or false. This is a known fact for specialists in cytology, as well as some differences in the behavior of the same cells in different passages.Vibrant™ CFDA SE dye, like any dye, has its drawbacks, which we indicated with reference [68]. It's published, so it's a fact. For example, the well-known dye trypan blue, recommended by ISO 10993-5 for determining cell viability, destroys stained cells, that overestimates viability results (trypan blue ruptures dead or dying immune cells leading to an over-estimation of cell viability) [Chan LL-Y, Kuksin D, Laverty DJ, Saldi S, Qiu J. Morphological observation and analysis using automated image cytometry for the comparison of trypan blue and fluorescence-based viability detection method. Cytotech. 2014;DOI 10.1007/s10616-014-9704-5]. Therefore, use of several staining methods for reliable and reasonable conclusions is recommended.This is what we did. Thus, the conclusion of respected Reviewer about the inappropriate choice of experimental methods is completely unfounded.

Question 2. The authors claimed they modified Fig 9b. But I don't think I can identify any changes in the revised manuscript v2.

Response: The Reviewer was very attentive in the first two rounds; therefore, it is impossible not to see that Figure 9 has a different bar colors and a different number of cells in Figure 9(b) in the panel of coated samples on day 7 (see below). We provided our explanations for Figure 9b in response to review-round 2 that the cell pictures in Figures 7b and 9b (day 7) do not visually contradict each other. New explanations in the text of the manuscript are not required, because Fig.9b only visualizes the data in Fig.9a. For persuasiveness, we showed in the review mode the previous and new versions of Figure 9 in the article.

Question 3. The scale bar in Fig 7 is clearly incorrect in some panels.

Response: It is not clear what dear Reviewer has in mind. The Zeiss microscope sets the scales automatically using own software. We remeasured once more the scales on all panels in Fig. 7, and they all correspond to 50 µm.

We hope that the Reviewer understands that instrumental measurements are much more accurate than visual sensations.
